# Detection of H5N1 High Pathogenicity Avian Influenza Viruses in Four Raptors and Two Geese in Japan in the Fall of 2022

**DOI:** 10.3390/v15091865

**Published:** 2023-09-01

**Authors:** Kei Nabeshima, Yoshihiro Takadate, Kosuke Soda, Takahiro Hiono, Norikazu Isoda, Yoshihiro Sakoda, Junki Mine, Kohtaro Miyazawa, Manabu Onuma, Yuko Uchida

**Affiliations:** 1Biodiversity Division, Ecological Risk Assessment and Control Section, National Institute for Environmental Studies, 16-2 Onogawa, Tsukuba 305-8506, Ibaraki, Japan; nabeshima.kei@nies.go.jp; 2Emerging Virus Group, Division of Zoonosis Research, National Institute of Animal Health, 3-1-5 Kannondai, Tsukuba 305-0856, Ibaraki, Japan; takadatey851@affrc.go.jp (Y.T.); minejun84032@affrc.go.jp (J.M.); miyazawak@affrc.go.jp (K.M.); 3Avian Zoonosis Research Center, Faculty of Agriculture, Tottori University, Tottori 680-8553, Tottori, Japan; soda@tottori-u.ac.jp; 4Laboratory of Microbiology, Department of Disease Control, Faculty of Veterinary Medicine, Hokkaido University, Sapporo 060-0818, Hokkaido, Japan; hiono@vetmed.hokudai.ac.jp (T.H.); nisoda@vetmed.hokudai.ac.jp (N.I.); sakoda@vetmed.hokudai.ac.jp (Y.S.)

**Keywords:** high pathogenicity avian influenza virus, H5N1, Japan, 2022/2023 season, peregrine falcon, greater white-fronted goose, eastern buzzard, phylogenetic analysis

## Abstract

In the fall of 2022, high pathogenicity avian influenza viruses (HPAIVs) were detected from raptors and geese in Japan, a month earlier than in past years, indicating a shift in detection patterns. In this study, we conducted a phylogenetic analysis on H5N1 HPAIVs detected from six wild birds during the 2022/2023 season to determine their genetic origins. Our findings revealed that these HPAIVs belong to the G2 group within clade 2.3.4.4b, with all isolates classified into three subgroups: G2b, G2d, and G2c. The genetic background of the G2b virus (a peregrine falcon-derived strain) and G2d viruses (two raptors and two geese-derived strains) were the same as those detected in Japan in the 2021/2022 season. Since no HPAI cases were reported in Japan during the summer of 2022, it is probable that migratory birds reintroduced the G2b and G2d viruses. Conversely, the G2c virus (a raptor-derived strain) was first recognized in Japan in the fall of 2022. This strain might share a common ancestor with HPAIVs from Asia and West Siberia observed in the 2021/2022 season. The early migration of waterfowl to Japan in the fall of 2022 could have facilitated the early invasion of HPAIVs.

## 1. Introduction

The H5 subtype of the high pathogenicity avian influenza virus (H5 HPAIV), originating from the H5N1 HPAIV isolated in southern China in 1996, has significantly impacted both wild birds and poultry in recent years [1]. The WHO/OIE(WOAH)/FAO H5N1 Evolution Working Group has classified H5 HPAIVs into 10 distinct clades, ranging from 0 to 9, and various subclades. Among these, clade 2.3.4.4 is the most prevalent worldwide [2]. A phylogenetic analysis of the hemagglutinin (HA) gene demonstrated that the H5N8 and H5N1 HPAIVs found in poultry and wild birds between October 2021 and May 2022 (during the 2021/2022 season) belonged to the G2 group of clade 2.3.4.4b [3,4,5,6]. In Japan, virus detection patterns have been strongly linked to wild bird migration during the winter months, suggesting that these migratory birds introduce the viruses to Japan [5]. Recent studies have uncovered genetic reassortments of H5 HPAIVs with low pathogenicity avian influenza viruses and other avian influenza viruses in Europe, North America, and South America [7], raising the potential for the emergence of new H5 HPAIV strains. This elevates the necessity for ongoing genetic analysis of these viruses.

In Japan, HPAIVs are typically found in the feces of the *Anatidae* family and in the lake and marsh waters where they gather for wintering, starting in late October, during the early stages of their migration [8]. Active surveillance is employed first, followed by passive surveillance of weakened or deceased wild birds. The viruses are later detected in the carcasses of *Anatidae*, and then in raptors, crows, and other resident birds. During the last two seasons in Japan (2020/2021: between October 2020 and May 2021, and 2021/2022), HPAIVs were detected in both poultry and wild birds [9,10]. The first detection occurred in the feces of wild birds on 24 October 2020, and in raptors on 4 December 2020, during the 2020/2021 season. Similarly, HPAIV was detected in environmental water on 8 November 2021, and in a raptor on 2 January 2022, following confirmed bird migration to the south in the 2021/2022 season. However, in the 2022/2023 season, HPAIVs were detected from the feces of *Anatidae* and swabs of raptors and geese between late September and mid-October, earlier than the peak of duck migration [10,11]. These occurrences are unlike any previously recorded. The HPAI epidemic in 2022/2023 was the largest ever recorded in Japan [9,10,12,13], with 84 outbreaks across 26 prefectures reported on poultry farms, and 242 cases across 28 prefectures in wild birds. During this season, we detected H5N1 and H5N2 viruses of clade 2.3.4.4b from poultry and H5N1, H5N2, and H5N8 viruses from wild birds [13,14].

In this study, we conducted a genetic analysis on the sequences of H5N1 HPAIVs detected in wild birds, i.e., three peregrine falcons (*Falco peregrinus*), one eastern buzzard (*Buteo japonicus*), and two greater white-fronted geese (*Anser albifrons*) (Table 1), during the early period of this season to determine their relationship with viruses detected in previous years.

## 2. Materials and Methods

### 2.1. Diagnosis of Specimens from Wild Bird Cases

Tracheal and/or cloacal swab samples were collected from weakened or deceased wild birds at the locations and dates specified in Table 1. These samples were tested for type A and B influenza viruses using a commercial rapid detection kit (ESPLINE Influenza A & B-N; Fujirebio Holdings, Inc., Tokyo, Japan) and further employed for sequencing, definitive subtype identification of influenza A virus, and verification of nucleotide sequences encoding multiple basic amino acids in the HA gene at the laboratory of the National Institute for Environmental Studies (NIES). The RNA extraction process followed the method outlined by Onuma et al. [8]. The H5 HA was examined using RT-qPCR with AgPath-ID One-Step RT-PCR Reagents (Thermo Fisher Scientific, Waltham, MA, USA) on a StepOnePlus Real-Time PCR System, as detailed by Heine et al. [15]. All gene segment amplifications were conducted using one-step RT-PCR with Uni12 and Uni13 primers, according to the procedure described by Hoffmann et al. [16]. These one-step RT-PCR products were used for H5 HA and N1 neuraminidase (NA) gene-specific PCR [17]. The PCR products of H5 HA and N1 NA were then electrophoresed, visualized, and purified with E-Gel SizeSelect Agarose Gels (Thermo Fisher Scientific, Waltham, MA, USA). These purified products were directly sequenced using a BigDye Terminator Cycle Sequencing Ready Reaction kit and a Genetic Analyzer model 3130 (Thermo Fisher Scientific, Waltham, MA, USA). The resulting fragment data were compiled using BioEdit version 7.2.5 [18]. Subsequently, a BLAST search of these sequences was performed to confirm the subtypes and the detection of multiple basic amino acids in HA was used to estimate pathogenicity [19].

### 2.2. Sequencing

HPAIVs were isolated from samples NIES208, NIES209, NIES214, NIES231, and NIES260 using the 9–11 day-old embryonated chicken egg inoculation method. Despite efforts to isolate the virus from sample NIES212, no infectious viruses were obtained from this specimen. The names of the six HPAIVs isolated or detected in this study are shown in Table 1. Nucleotide sequences for the PFI/NIES208 and PFD/NIES231 samples were acquired using MiSeq (Illumina, Sandiego, CA, USA) at the National Institute of Animal Health, following the procedure previously described [4]. The full-length viral genomes of GWFD/NIES209 and EBI/NIES260 were amplified via RT-PCR, employing the primer sets outlined by Ip et al. [20]. Oxford Nanopore libraries were prepared with the NEB Ultra II End Repair/dA-Tailing Module (New England Biolabs, Ipswich, MA, USA) and sequenced on a Flongle (Oxford Nanopore, Oxford, UK) using a Nanopore Direct cDNA sequencing kit (Nanopore, Oxford, UK). The read data were then mapped and assembled with FluGAS version 2 (World Fusion, Tokyo, Japan). Specific gene sequences were tentatively determined as follows: polymerase basic protein 2 (PB2), polymerase acidic protein (PA), nucleoprotein (NP), and non-structural protein (NS) genes of PFD/NIES212, and PB2, PA, NP, matrix protein (MP), and NS genes of GWFD/NIES214 were identified using PCR and Nanopore Flongle with IRMA [21]. The polymerase basic protein 1 (PB1) sequence of GWFD/NIES214 was discerned by PCR and a BigDye Terminator Cycle Sequencing Ready Reaction kit. However, the PB1 and MP gene sequences of PFD/NIES212 could not be determined due to a lack of samples. All gene fragment sequences ascertained in this study have been registered in the GenBank/EMBL/DDBJ and the Global Initiative on Sharing All Influenza Data (GISAID) database (http://platform.gisaid.org (accessed on 31 August 2023)) (Table 1).

### 2.3. Phylogenetic Analysis

Utilizing a BLAST search on the GISAID databases, we downloaded nucleotide sequences that displayed high similarity to our HPAIV sequences. Specifically, we obtained the top 250 BLAST hits for the HA and NA genes, and the top 100 BLAST hits for the PB1, PB2, PA, NP, M, and NS genes. These sequences were aligned with the HPAIV sequences using BioEdit (version 7.2.5), and any ambiguous or duplicate sequences were subsequently removed. The sequence counts used for phylogenetic analyses were as follows: HA, 653 sequences; NA, 661; PB2, 286; PB1, 205; PA, 232; NP, 242; MP, 237; and NS, 260. We constructed maximum likelihood trees based on the aligned sequences with FastTree (version 2.1.10) [22], utilizing a generalized time-reversible model and 1000 resampling iterations for statistical support. The resultant phylogenetic tree was visualized using FigTree (versions 1.4.4 and 1.3.1) [23].

## 3. Results and Discussion

Six HPAIV strains, isolated or detected from four raptors and two geese in the early period of the 2022/2023 season, were sequenced and analyzed to ascertain their genetic origins (Table 1). A definitive diagnosis, employing both the multiple basic amino acids of the HA protein and a BLAST search, revealed that the six strains from wild birds were H5N1 HPAIVs. Analysis of the HA genes’ phylogenetic tree revealed that all these strains belong to the G2 genetic group of clade 2.3.4.4b [6]. Additionally, they were classified into three subgroups, specifically G2b, G2d, and G2c, within the same clade (Figure 1a). PFI/NIES208 and PFD/NIES212 were classified into groups G2b and G2c, respectively (Figure 1b,d), while the remaining four strains were classified as G2d (Figure 1c). Phylogenetic analyses of NA and internal genes (excluding PB1 and M genes) confirmed that the six isolates were divided into three subgroups, as these isolates clustered in the HA phylogenetic tree (Figure 2, Appendix A). Though detailed origins of each strain are provided later, it is notable that G2d and G2b included Japanese HPAIVs that caused poultry outbreaks in the 2021/2022 season (blue-colored strains), except for G2c (Figure 1b,c).

PFI/NIES208 was classified as G2b [3] (previously known as 20E [4] and group 2 [5]) (Figure 1b). HPAIVs classified as G2b shared common ancestors with H5N8 isolates in Europe during the 2020/2021 season [4]. As shown in Figure 1b, PFI/NIES208 was found to be closely related to strains isolated from Eurasian wigeons (*Mareca penelope*) in Japan in October 2022 (red-colored strains), and other isolates from wild birds and poultry in Japan (Kagoshima, Ehime, Aomori, Hyogo, Miyazaki, Saitama, and Chiba prefectures) during the 2021/2022 season (blue-colored strains). The G2b group also encompassed HPAIV strains from poultry and wild birds in China, Korea, and Indonesia between October 2021 to April 2022. As displayed in Figure 1, Figure 2 and Appendix A, phylogenetic analyses of HA, NA, and internal genes affirmed that the genotype of PFD/NIES208 was congruent with the G2b viruses of the 2021/2022 season (e.g., A/chicken/Kagoshima/21A6T/2021 (H5N1) [3]).

The strains GWFD/NIES209, GWFD/NIES214, PFD/NIES231, and EBI/NIES260 were classified as G2d [3] (Figure 1c), previously defined as 21E [4] and group 1 [5]. They share common precursors with H5N1 isolates in Europe during the 2021/2022 season [4]. Previous studies have also estimated that the G2d group viruses entered Japan from Africa via Eurasia [5,24]. These were isolated from poultry and wild birds during 2021/2022 (blue-colored strains) in Hokkaido Prefecture (Figure 1c), as reported previously [5]. Additionally, the NA and internal genes of these four strains were closely related to H5N1 HPAIVs from 2021/2022 in Hokkaido Prefecture (Figure 2 and Appendix A), indicating a consistent genetic background. The G2d group also included strains from the United States, Canada, and Russia between February 2022 and January 2023 (Figure 1c).

Strain PFD/NIES212 was classified within the G2c group [3], previously referred to as 21RC [13]. The HA genes of the G2c viruses are closely related to those of H5N1 HPAIVs detected in ducks and chickens in Korea and Russia from August to November 2022 (Figure 1d). They also shared similarities with H5N1 viruses found in ducks, geese, swans, chickens, and turkeys across Korea, China, Russia, and Bangladesh from October 2021 to March 2022 (Figure 1d). These results hint at the possible origin of the G2c group viruses in East and South Asia and West Siberia. Since the first detection of the G2c virus (PFD/NIES212) in Japan in October 2022, many G2c viruses were isolated from mallard ducks, swans, raptors, crows, and chickens until April 2023. The phylogenetic analysis, excluding the PB1 and MP genes, showed that six viral genes of PFD/NIES212 were similar to Asian H5N1 HPAIV strains in 2021/2022, such as A/goose/Hunan/SE284/2022 (H5N1) (SE284) (Figure 1, Figure 2 and Appendix A). This suggests the introduction of an SE284-like virus into Japan in the 2022/2023 season. Notably, PB2 of this virus also had similarities with H10, H6, H11, and H3 subtype avian influenza viruses, and H5 low pathogenicity avian influenza viruses found in wild birds (Appendix A).

Mallard ducks (*Anas platyrhynchos*), Eurasian spot-billed ducks (*Anas zonorhyncha*), and northern pintails (*Anas acuta*) were reported to play an important role in carrying avian influenza viruses into Japan [8]. According to migratory bird census data from Yatsu-higata, or Yatsu tidal flat, Chiba Prefecture—the nearest census site where the falcon carrying the G2b group virus was discovered—northern pintails were observed a month earlier in 2022 (first observed on 23 September) compared to 2020 and 2021 (first observed on 25 October 2020, and 17 October 2021, respectively) (Appendix A) [25]. These observations suggest a possible shift in migration timing for northern pintails and other waterfowl, leading to earlier virus detection compared to previous years. H5N1 HPAIV was found in northern pintails’ feces in Hokkaido, Japan, on October 8 [14]. Falcons might have become infected with HPAIV by preying on infected birds like northern pintails in the vicinity. Supporting this notion is the absence of HPAIV reports in wild birds and poultry in Japan during the summer period of 2022 (from late May to mid-September) [10]. Consequently, it seems improbable that viruses like the G2b and G2d groups, detected the prior year, lingered in Japan, and they likely were reintroduced by migratory birds in the fall of 2022. For a clearer understanding of the G2b and G2d viruses’ origins detected in the 2022/2023 season, comprehensive genetic information on HPAIVs from various countries is required.

## Figures and Tables

**Figure 1 viruses-15-01865-f001:**
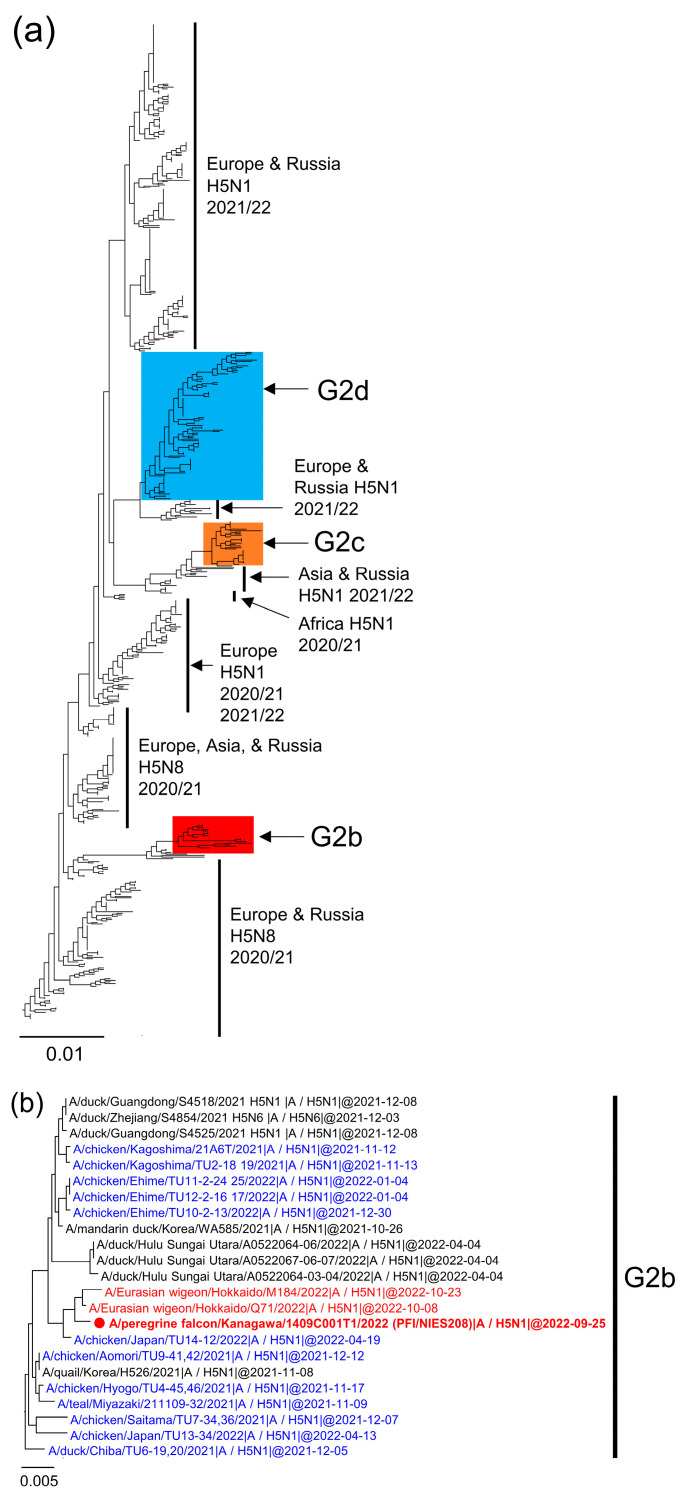
Phylogenetic tree of the hemagglutinin (HA) gene segment. Panel (**a**) displays the complete phylogenetic tree of the HA genes. Panels (**b**–**d**) are detailed views of the phylogenetic trees focusing on the HA genes of the G2b, G2d, and G2c subgroups, respectively. Strains isolated or detected in Japan during the 2022/2023 season are represented in red, while those from the 2021/2022 season are in blue. Strains marked with filled circles indicate viruses isolated or detected in this study.

**Figure 2 viruses-15-01865-f002:**
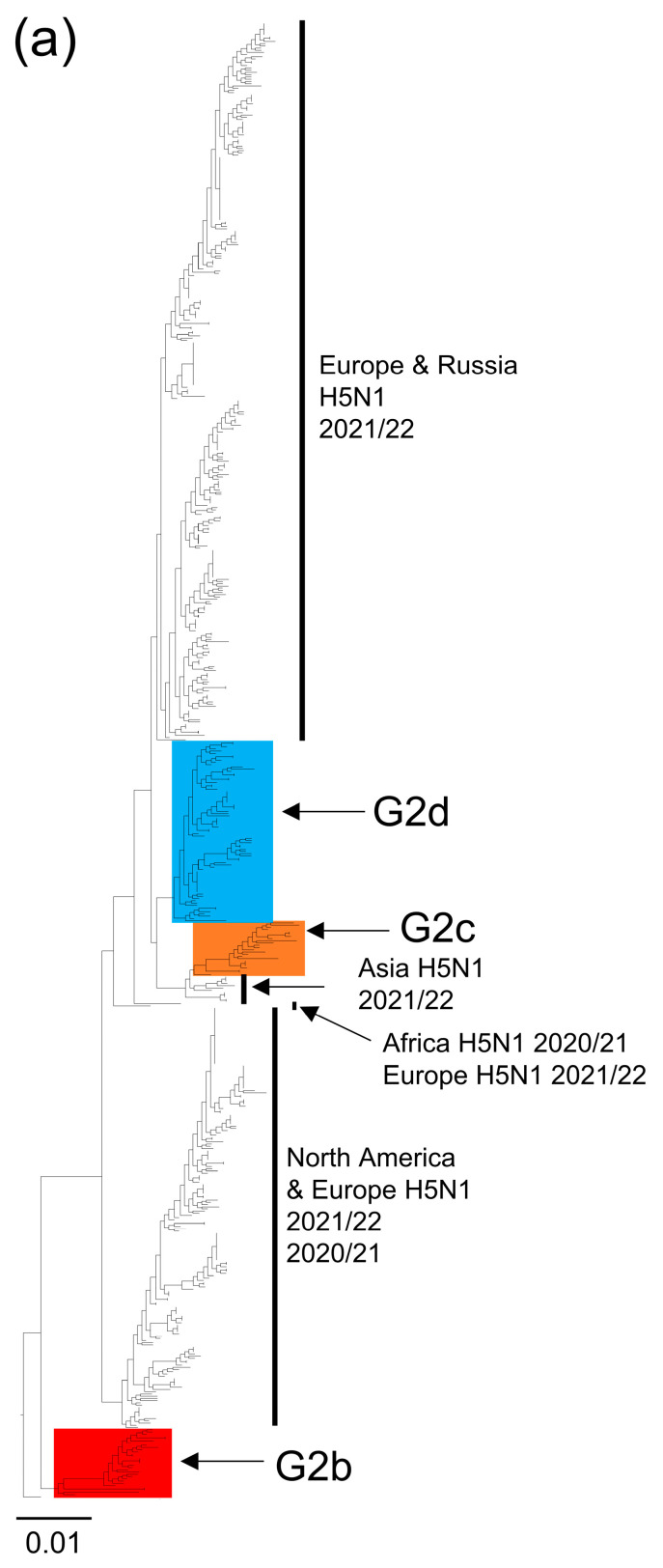
Phylogenetic tree of the neuraminidase (NA) gene segment. Panel (**a**) displays the complete phylogenetic tree of the NA genes. Panels (**b**–**d**) are detailed views of the phylogenetic trees focusing on the NA genes of the G2b, G2d, and G2c subgroups, respectively. Strains isolated or detected in Japan during the 2022/2023 season are represented in red, while those from the 2021/2022 season are in blue. Strains marked with filled circles indicate viruses isolated or detected in this study.

**Table 1 viruses-15-01865-t001:** Sample information.

**Sample ID**	**Host Species**	**Prefecture**	**Date**	**Results of Virus Isolation**	**Strain Name (Abbreviation)**	**Accession Number**
**NIES208**	Peregrine Falcon (*Falco peregrinus*)	Kanagawa	25 September	+	A/peregrine falcon/Kanagawa/1409C001T1/2022 (PFI/NIES208)	EPI2223228-33 (GISAID)
**NIES209**	Greater white-fronted goose (*Anser albifrons*)	Miyagi	4 October	+	A/greater white-fronted goose/Miyagi/0410D001/2022 (GWFD/NIES209)	EPI2197765-72 (GISAID)
**NIES212**	Peregrine Falcon	Fukui	11 October	-	A/peregrine falcon/Fukui/NIES212/2022 (PFD/NIES212)	LC762442, LC762448, LC774734, LC774738, LC774739, LC774741 (GenBank)
**NIES214**	Greater white-fronted goose	Miyagi	14 October	+	A/greater white-fronted goose/Miyagi/NIES214/2022 (GWFD/NIES214)	LC762443, LC762449, LC774733, LC774735-7, LC774740, LC774742 (GenBank)
**NIES231**	Peregrine Falcon	Niigata	16 October	+	A/peregrine falcon/Niigata/1510C001T/2022 (PFD/NIES231)	EPI2632285-92 (GISAID)
**NIES260**	Eastern Buzzard (*Buteo japonicus*)	Niigata	21 October	+	A/eastern buzzard/Niigata/1501B001/2022 (EBI/NIES260)	EPI2318005-12 (GISAID)

## Data Availability

Data supporting this study are available from the corresponding author upon reasonable request.

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
