# Peer review of "Detection of H5N1 High Pathogenicity Avian Influenza Viruses in Four Raptors and Two Geese in Japan in the Fall of 2022"

_viruses, 2023, doi:10.3390/v15091865_

Round 1

Reviewer 1 Report

Thank you for the brief report "Detection of H5N1 HPAI viruses in four raptors and two geese in the fall of 2022." Perhaps you could add "in Japan" to your title to make it more prescript.

I have most of my questions about your materials and methods.

1. Why was a rapid detection kit used instead of using a regular real-time PCR targeting matrix gene for AIV identification? I would think a PCR step would be much more beneficial to look at odds of isolating a virus +/- generating a strong enough sequence for next generation sequencing based on ct values. 

2. What is the actual "sample" you used for sequencing? Did you do direct sequencing using Nanopore from the swabs (clinical sample), or did you sequence the virus isolated from the sample? Either way, it would be best if these could be compared, especially in the case of the Peregrine falcon sample where virus was not isolated. Also, I would add that information and what sequencing platform you used into your abstract (word limit permitting). 

3. Please provide the catalog number for the direct cDNA sequencing kit - is this still available or retired from Nanopore? Please provide justification on why this sequencing kit/barcoding kit was used compared to the PCR-based cDNA kits or other kits. 

L73 - weakened or deceased wild birds... did they have clinical signs of HPAI? Were other causes of death/clinical signs examined in these birds? Why were these birds selected for sampling/sequencing?

L117 - could you clarify what you mean by "lack of samples?" I think you meant to say perhaps low number of viral reads or "lack of adequate viral RNA?" I realize that the direct cDNA kit requires a higher input for analysis so perhaps you didn't have enough host RNA on the swabs??

The intent/goal of the paper is unclear to me. I do not think you can make any sort of broad conclusion based on 6 sequences, but there will be more impact if these could be compared to the timeline of some of the introduction into domestic poultry in Japan or other parts of Asia as part of the discussion. You alluded to that in your first paragraph of the discussion, but I would like to see what kind of passive surveillance program for HPAI exists in Japan and why these specific wild bird samples were collected (e.g. targeted sampling? found dead close to a cluster outbreak? etc...)

Reviewer 2 Report

The manuscript is well written, I haven't any comment at all . All is perfect for me. Best wishes

Author Response

The manuscript is well written, I haven't any comment at all . All is perfect for me. Best wishes

Thank you so much for your comment.

Reviewer 3 Report

The authors analyzed H5N1 avian influenza viruses collected in Japan in the fall of 2022. Detailed analysis based on the genetic analysis allowed to investigate the origin of the viruses. I think this is well written, and suitable for publication in this journal.

There are some comments as follows.

Specific Remarks:

1) P.2 line 80: The real-time PCR assay detects HA gene. I feel that the expression The presence of H5 hemagglutinin” is strange. In addition, HA is already abbreviated on line 78.

2) P.2 line 87: Please add the manufacture name after “E-Gel SizeSelect Agarose Gel”.

3) P.8 line 158-159: I think this expression is misleading. In Figure 1b, blue-colored strains were collected not only in Kagoshima and Ehime prefectures, but also in Aomori and other prefectures.

4) P.9 line 171-173: The authors concluded that G2d group viruses were reintroduced to Japan in the 2022/2023 season. The NA and internal genes of the four G2d viruses analyzed in this study should be compared not only with domestic viruses but also with viruses collected in other countries. (The G2b group virus was being compared.)
